# Gesture Recognition Based on Multiscale Singular Value Entropy and Deep Belief Network

**DOI:** 10.3390/s21010119

**Published:** 2020-12-27

**Authors:** Wenguo Li, Zhizeng Luo, Yan Jin, Xugang Xi

**Affiliations:** 1Institute of Intelligent Control and Robotics, Hangzhou Dianzi University, Hangzhou 310018, China; wgli_hz@163.com (W.L.); xixi@hdu.edu.cn (X.X.); 2Security Department, Hangzhou Dianzi University, Hangzhou 310018, China; jinyan@hdu.edu.cn

**Keywords:** surface electromyography, S-transform, multiscale singular value decomposition, permutation entropy, deep belief network, gesture recognition

## Abstract

As an important research direction of human–computer interaction technology, gesture recognition is the key to realizing sign language translation. To improve the accuracy of gesture recognition, a new gesture recognition method based on four channel surface electromyography (sEMG) signals is proposed. First, the S-transform is applied to four channel sEMG signals to enhance the time-frequency detail characteristics of the signals. Then, multiscale singular value decomposition is applied to the multiple time-frequency matrix output of S-transform to obtain the time-frequency joint features with better robustness. The corresponding singular value permutation entropy is calculated as the eigenvalue to effectively reduce the dimension of multiple eigenvectors. The gesture features are used as input into the deep belief network for classification, and nine kinds of gestures are recognized with an average accuracy of 93.33%. Experimental results show that the multiscale singular value permutation entropy feature is especially suitable for the pattern classification of the deep belief network.

## 1. Introduction

Gesture recognition, which uses a computer to convert hand movement information into a specific target application, has become a research focus in the field of human–computer interaction and rehabilitation medicine [1,2]. Sign language (SL) is an action set composed of many gestures that have meaning. Therefore, gesture recognition is the key to SL translation. SL translation contributes to the smooth communication between the deaf/mute and normal hearing/speaking people and greatly improves the deaf/mute’s social participation, which will bring gospel to many deaf/mute communities around the world.

Surface electromyography (sEMG) is a kind of bioelectrical signal [3,4,5] that reflects the neuromuscular system activity collected from the muscle surface and contains abundant information related to gesture action. As sEMG signal acquisition is simple and non-invasive, gesture recognition based on sEMG [6,7] is favored by an increasing number of researchers at home and abroad. Gesture recognition is the main approach to achieving SL translation. At present, more than 5300 Chinese SL words exist [8,9], and hundreds of common SL words are used in daily life. Recognizing SL words individually and in isolation will require heavy training and calculation. In fact, SL actions can be decomposed into several standardized gestures and movement trajectories, and the translation of SL words will be transformed into the recognition of several specific gestures. An sEMG signal is produced with the contraction of muscle and can be obtained as long as the related muscle is sound. Different gesture actions need the cooperation of different muscles to drive the action. Therefore, the collected sEMG signal contains spatial pattern information related to the muscle position. The corresponding gesture can be recognized by analyzing the pattern information of multiple muscles. Yang et al. [9] realized the recognition of eight gestures (corresponding to 150 words) using four channel sEMG signals, with an average accuracy of 87.02%. Zhang et al. [10] used four forearm and one palmar sEMG signals to classify 18 independent Chinese SL words, reaching an accuracy of 91.4%. Zhang et al. [11] used five channel sEMG signals to classify 72 words, with an accuracy of 90.2%. Increasing the layout of sensors improves the recognition accuracy, but also increases the complexity of the recognition system. This paper attempts to analyze four channel sEMG signals to recognize nine standardized gestures.

The sEMG signal is weak, time-varying, and non-stationary. Performing feature analysis on signals is the key to improving the recognition accuracy. Han et al. [12] used a short-time Fourier transform (STFT) to analyze sEMG signals, effectively improving the recognition rate of action. In [13,14], wavelet transform was used to decompose sEMG signals into multiscale and achieved good experimental results. Xu et al. [15] used the Wigner Ville distribution to analyze sEMG signals to predict the knee joint torque. Time-frequency analysis methods, such as STFT, wavelet transform, and the Wigner Ville distribution, cannot achieve high resolution in the time and frequency domains at the same time [16], so the detailed features of signals cannot be effectively enhanced. However, S-transform (ST) is the improvement of the STFT and wavelet transform, which overcomes the defect of the STFT window function and effectively improves the resolution of the time-frequency feature [17,18]. ST exhibits excellent performance in analyzing time-varying signals. The time-frequency matrix (TFM) output of ST contains rich time-frequency information. TFM’s singular value effectively reflects the global characteristics of the matrix but contains more redundant information, so the details and local characteristics are insufficient [19,20,21]. Therefore, the TFM is locally divided along the time and frequency axes, and the multiscale singular value decomposition (SVD) of the global and local sub matrices are performed separately, which has a positive effect on feature extraction.

To further realize the feature transformation of multiscale singular values while effectively reducing the dimensions of multiple eigenvectors, the entropy of multiscale singular values of multiple muscles is calculated. Given the limitations of weak anti-noise ability and complex data preprocessing, approximate entropy and sample entropy can hardly extract the detailed characteristics of the sEMG signal itself effectively [22]. Permutation entropy (PE) takes the change in sorting mode as an important feature of time series, which has the advantages of fast calculation and strong anti-noise ability. Given the difference between the number of motor units involved in muscle activities and the nerve conduction velocity of action potential under different gesture actions, the complexity of sEMG signal is different. The vector volatility information reflected by PE is a complexity measure of the vector volatility model. Combined with the matrix singular values arranged in descending order, PE is selected to reflect the intrinsic complex characteristics of the sEMG signal.

The multiscale analysis method for the time-frequency features of multiple muscles ensures the integrity of the feature information of sEMG signals, but it greatly increases the design difficulty and training load of the gesture recognition classifier. A deep belief network [23,24,25,26] (DBN) is a deep learning model with an excellent feature learning ability. A DBN is composed of several restricted Boltzmann machines (RBMs), which can alleviate the problem of multi-layer neural network training. PE and RBMs are probability distribution models that facilitate the full absorption and learning of PE features by the DBN compared with other methods, making the PE features suitable for the pattern classification of the DBN. Figure 1 is the framework of gesture recognition methods designed in this paper. In the paper, the four channel sEMG signals are collected and then transformed by ST. The multiple TFM output of ST is processed by multiscale SVD to obtain the time-frequency joint features with strong robustness. The multiscale singular value PE eigenvalues of multiple muscles are calculated as the feature vector and used as input into the DBN for classification. Finally, nine kinds of gestures are recognized. The results show that the combination of sEMG signal multiscale singular value PE and the DBN can achieve better recognition accuracy of 93.33% without increasing the number of sEMG channels and signal types. This method is effective for improving the accuracy of gesture recognition. The comparison of methods is shown in Table 1, it can be found that this method is superior to other comparison methods. Due to the non-stationary characteristics of sEMG signal, multiscale singular value PE analysis is carried out. The multiscale signal features make the classification more accurate, and the PE feature of sEMG signal is easier to fully absorb and learn by the DBN, so better recognition accuracy is achieved.

## 2. Materials and Methods

### 2.1. Muscle Selection

In this paper, the Trigno Wireless sEMG acquisition system (Delsys Ltd., Boston, MA, USA) is used to collect multichannel sEMG signals of forearm muscles. The execution of different gestures is driven by specific muscles, so the selection of target muscle is related to the accuracy of gesture recognition. The muscles studied in [9,10,11] are mainly extensor carpi radialis (ECR), extensor carpi ulnaris, extensor digitorum (ED) and palmaris longus. Through the decomposition of SL gestures, we can see that there are many kinds of finger movements in gestures. Therefore, it is necessary to increase the muscles related to finger movements in the analysis process. Thus, four groups of muscles, namely ECR, ED, Flexor digitorum superficialis (FDS) and extensor pollicis brevis (EPB), were selected as signal acquisition objects. The sEMG signal acquisition sensors are arranged at the positions of these four groups of muscles, and the specific positions are shown in Figure 2.

### 2.2. Gesture Category

According to the analysis and induction of Chinese SL [8], the gestures of common SL words are decomposed into nine standardized gestures, as shown in Figure 3. The specific gesture description is shown in Table 2.

### 2.3. ST of sEMG Signal

ST is a reversible time-frequency analysis method proposed by R.G. Stockwell [27,28]. Its basic idea is to add a Gaussian window whose width is inversely proportional to frequency, and then Fourier transform the signal.

For a continuous time signal h(τ), its ST of S(τ,f) is defined as [29,30]:(1)$$S(\tau ,f)=\int_{-\infty }^{\infty }h(t)w(\tau -t,f)e^{-i2\pi ft}dt$$
where, w(t,f) is the Gaussian window function and τ is the translation factor controlling the position of the Gaussian window on the time axis.

Let R(α,f) be the Fourier transform of S(τ,f) from time τ for frequency α. According to the convolution theorem, we get the following formula:(2)$$R(\alpha ,f)=H(\alpha +f)e^{-2\pi^{2}\alpha^{2}/f^{2}},(f\neq 0)$$

When τ→jT (*T* is the sampling period and *N* is the data length), the discrete form of ST can be obtained as follows:(3)$$\left\{ \begin{array}{l} S\left[jT,\frac{n}{NT}\right]=\sum_{m=0}^{N-1}H\left[\frac{m+n}{NT}\right]e^{\frac{-2\pi^{2}m^{2}}{n^{2}}}e^{\frac{i2\pi mj}{n^{2}}},(n\neq 0)\\ S\left[jT,0\right]=\frac{1}{N}\sum_{m=0}^{N-1}h\left[\frac{m}{NT}\right],(n=0) \end{array}\right.$$

Therefore, ST can be regarded as a STFT with a variable window function, which has variable time-frequency resolution, and can meet the analysis requirements of different frequency domain signals. The output TFM of ST is a complex matrix, so the TFM mentioned below is the matrix after modular operation. Figure 4 is a comparison of the results of the sEMG signal processed by the ST and the STFT, respectively. Considering that the main frequency bands of sEMG are distributed in the range of 10~500 Hz [14,31], the spectrum analysis range of ST and STFT is 0~500 Hz, as shown in Figure 4b,c; it is obvious that the resolution of ST is better than that of STFT. The TFM diagram of ST and STFT are shown in Figure 4d,e respectively. The abscissa of the matrix is the time vector, which represents the change of signal amplitude with frequency under a certain time; the ordinate is the frequency vector, which represents the change of signal amplitude with time under a certain frequency; the amplitude intensity is represented by color depth. Figure 4e shows that the areas outside the frequency point of 300 Hz are filtered out relatively cleanly (including useful signals), which is not conducive to the analysis of the detailed characteristics of signals.

Figure 5 is a 3D time-frequency-amplitude diagram of the signal in Figure 4a after the ST and STFT operation, which is more convenient for analysis and observation. It can be clearly found that the time-frequency resolution of ST is greatly improved compared with STFT, which is more conducive to the analysis of the signal details.

### 2.4. Multiscale Singular Value PE

SVD is an important matrix decomposition method, which is widely used in signal feature extraction. The singular value not only contains important matrix characteristic information, but also is insensitive to the disturbance of matrix elements [19], and has relative stability in feature extraction. Therefore, SVD is applied to the TFM, and its singular value characteristics are analyzed.

In SVD theory, any matrix A of m×n order can be decomposed into:(4)$$A=U\Lambda V^{T}$$
where U and V are orthogonal matrices of m×m and n×n order, respectively, and $\Lambda =diag(\lambda_{1},\lambda_{2},\cdots ,\lambda_{k})$. (where k=rank(A)) is a diagonal matrix. Its diagonal elements are singular values of matrix A which arranged in descending order.

Since Λ is a diagonal matrix, matrix A of m×n order with rank *k* can be expressed as the sum of *k* sub matrices of m×n order with rank 1.
(5)$$A=U\Lambda V^{T}=\sum_{i=1}^{k}\lambda_{i}u_{i}v_{i}=\sum_{i=1}^{k}\lambda_{i}A_{i}$$
where $u_{i},v_{i}$ is the column i singular value vector of *U* and *V* respectively, and $A_{i}$ is the sub matrix containing $u_{i}$ and $v_{i}$. In practical application, the matrix A represents the time-frequency information of the sEMG signal, the corresponding $u_{i}$ and $v_{i}$ represent the time and frequency information, respectively, and the size of the singular value represents the amount of information in the time-frequency range. The matrix is decomposed into a series of time-frequency subspaces corresponding to singular values and singular value vectors, and the signal feature types can be distinguished by judging the size of singular values.

SVD is applied to the whole matrix, and the obtained singular values reflect the global characteristics of the matrix, so the detailed and local characteristics are not sufficiently described [20,21]. In order to extract signal features more comprehensively and effectively, this paper proposes a signal feature extraction method of local SVD as a supplement. Firstly, the whole TFM is divided into a sub matrix along the time axis and frequency axis. Then, SVD is applied to each sub matrix. Since the original matrix is locally divided, more effective detail features can be obtained. The specific steps are as follows:
(1)The TFM A is obtained by ST of the sEMG signal.(2)Calculate the global singular value eigenvector $\lambda_{A}=[\lambda_{1},\lambda_{2},\cdots ,\lambda_{k}]$ of matrix A.(3)A is divided into q sub matrices along the time axis and p sub matrices along the frequency axis. The division method is shown in Figure 6.(4)SVD is applied to the p+q sub matrix to obtain the corresponding singular value sequence.(5)Since the singular value sequence of each sub matrix decays rapidly in numerical value, the largest singular value of each sub matrix is selected to construct eigenvectors, that is $\lambda_{t}=[\lambda_{t\max,1},\lambda_{t\max,2},\cdots ,\lambda_{t\max,q}]$ and $\lambda_{f}=[\lambda_{f\max,1},\lambda_{f\max,2},\cdots ,\lambda_{f\max,p}]$.

Where $\lambda_{t\max,•}$ is the maximum singular value of q sub matrices divided along the time axis, and $\lambda_{f\max,•}$ is the maximum singular value of p sub matrices divided along the frequency axis.

(6)PE is performed on multiscale singular values. As a nonlinear dynamic method based on complexity measurement, PE has been gradually applied to the analysis of complex bioelectrical signals [32]. PE is mainly used to analyze the changes of nonlinear time series, and its basic principles are defined as follows:

An arbitrary time series $\left\{x_{1},x_{2},\cdots x_{l}\right\}$ is given, where l is the data length.

$X_{t}=[x_{t},x_{t+\tau },\cdots ,x_{l+(m-1)\tau }]$ is obtained by reconstructing its phase space, where m is the embedding dimension and τ is the delay time.

$X_{t}$ is arranged in descending order, and each vector in m dimensional space is mapped to one of all m! sorting patterns.

For any sequence π, let T denote the times it appears in time series analysis, then its relative probability distribution is as follows:(6)$$p_{i}(\pi )=\frac{T}{N-(m-1)\tau },i\leq m!$$

Then the definition of PE [30,31] is as follows:(7)$$H(m)=-\sum_{i=1}^{m!}p_{i}(\pi )\ln p_{i}(\pi )$$

The volatility information of the m-dimensional vector reflected by PE is actually a complexity measure of m dimensional vector volatility pattern.

The singular value entropy eigenvector is obtained by a PE operation on the singular values $\lambda_{t}$, $\lambda_{f}$ and $\lambda_{A}$ in the above steps, which is recorded as $F=[E_{t},E_{f},E_{A}]$. Where $E_{t}$ is the maximum singular value PE of the local division of the time axis, $E_{f}$ is the maximum singular value PE of local division of frequency axis, and $E_{A}$ is the global singular value PE of matrix A. Thus, the multiscale singular value entropy eigenvector is constructed from the global to the local, and the signal characteristics are described comprehensively.

### 2.5. Gesture Classification

The DBN was proposed by Geoffrey Hinton in 2006 [25], which promotes the rapid development of deep learning, and improves the generalization ability and adaptive ability of the training process. It has better performance than the shallow network structure in complex classification, and it has been widely used in image recognition, speech recognition and the biological signal processing fields [33,34].

The DBN is a multi-layer probabilistic machine learning model combining unsupervised learning and supervised learning. It is composed of a multi-layer unsupervised RBM and a one layer supervised classifier, as shown in Figure 7. In the first stage, the greedy unsupervised learning algorithm is used to initialize the parameters of deep network structure layer by layer. The output of the previous RBM is taken as the input of its higher RBM, and each RBM is trained from bottom to top.

The RBM is a probability distribution model based on energy. In Figure 7, the energy function (v,h), composed of a visible layer and a hidden layer is [35]:(8)$$E\left(v,h|\theta \right)=-\sum_{i=1}^{n}a_{i}v_{i}-\sum_{j=1}^{m}b_{i}h_{i}-\sum_{i=1}^{n}\sum_{j=1}^{m}w_{ij}v_{i}h_{j}$$
where $\theta =\left(w_{ij},a_{i},b_{j}\right)$ is the parameter of the RBM model, $a_{i}$ and $b_{j}$ are the offsets of visible cell $v_{i}$ and hidden cell $h_{j}$ respectively, $w_{ij}$ is the connection weight between visible cell $v_{i}$ and hidden cell $h_{j}$, and n and m are the number of visible cell $v_{i}$ and hidden cell $h_{j}$ respectively.

From Equation (8), we can get the joint probability distribution of the given (v,h) as follows:(9)$$P\left(v,h|\theta \right)=\frac{1}{Z(\theta )}\exp\left(-E\left(v,h|\theta \right)\right)$$
where $Z(\theta )=\sum_{v}\sum_{h}\exp\left(-E\left(v,h|\theta \right)\right)$ is the normalization coefficient.

The states of the hidden cells in the RBM are independent of each other. When visible $v_{i}$ is given, the probability of hidden cell $h_{i}$ being activated (set to 1) is as follows:(10)$$P\left(h_{j}=1|v,\theta \right)=sigmoid\left(b_{j}+\sum_{i=1}^{n}w_{ij}v_{i}\right)$$

Similarly, when the state of the hidden cell $h_{i}$ is determined, the probability of visible cell $v_{i}$ is activated is as follows:(11)$$P\left(v_{i}=1|h,\theta \right)=sigmoid\left(a_{i}+\sum_{j=1}^{m}w_{ij}h_{j}\right)$$
where sigmoid(•) is the activation function and its value will be mapped to the (0, 1) interval.

For the RBM model with a given number of visible and hidden cells, the parameter θ needs to be determined by training. The training goal is to make the reconstructed data of the RBM model consistent with the given training sample data as far as possible. Because the distribution function Z(θ) of RBM is difficult to calculate by the naive method, this paper uses Hinton’s contrastive divergence algorithm [36] to train the unsupervised RBM and to solve for the optimal value of θ.

In the second stage, the supervised algorithm is used to fine-tune the network parameters after initialization. The last layer of the DBN is set as the Softmax classifier, the weight obtained from pre-training is taken as the initial weight of the DBN network, and the whole model is fine-tuned from top to bottom. This learning method overcomes the problems encountered in deep network training, and makes the deep network learning more efficient.

## 3. Results and Discussion

### 3.1. Feature Extraction Experiment

Eight healthy volunteers (five males and three females) aged from 20 to 40 were recruited and trained with the nine kinds of gestures. During the experiment, each volunteer sat and performed these nine gestures, which were repeated five times from gesture 1 (FFE), gesture 2 (FFC) to gesture 9 (FTIF). The sEMG signals of ECR, ED, FDS and EPB muscles were simultaneously recorded through Trigno. A total of 40 groups of sEMG data of four muscles were collected. Among them, 20 groups were training samples and the other 20 groups were test samples. Figure 8 shows the sEMG signals of ECR, ED, FDS and EPB muscles during FFE gesture execution.

Four sets of TFM diagrams are obtained by ST of four channel sEMG signals, as shown in Figure 9. In the figure, the abscissa is the time vector, the ordinate is the frequency vector, and the amplitude intensity is represented by color depth.

Singular value implies the important information of the matrix. In this paper, multiscale analysis of the TFM output by ST can effectively obtain more detailed features of the matrix. SVD is applied to the four sets of TFMs in Figure 9 and the singular values are calculated. As shown in Figure 10c, the singular value sequence of the matrix decays rapidly in numerical value, so only the first 20 singular values are listed in the figure. The TFM is divided into 16 sub matrices along the time axis and the frequency axis respectively. Each sub matrix is SVD, and the maximum singular value of each sub matrix is taken as the eigenvalue, as shown in Figure 10a,b.

The entropy eigenvector of four muscles is constructed by PE operation on the multiscale singular value of Figure 10, and the vector dimension is greatly reduced. The multiscale singular value PE of 20 groups of training samples were calculated, and the characteristic distribution was represented by a scatter plot. Figure 11a–d shows the multiscale singular value PE eigenvalues of ECR, ED, FDS and EPB muscles, respectively. In these figures, the x-, y- and *z*-axis represent the local singular value PE of time axis division, local singular value PE of frequency axis division and global singular value PE, respectively.

Through observation and analysis, the features of gestures FFE, FFC, EIMF ETIF and FTIF in Figure 11a are obviously different. The distance between classes of gestures ET and FT in Figure 11c is large, and gestures EIF and ETP can also be obviously classified in Figure 11b. Then, with the aid of the features in Figure 11d, the classification of the nine kinds of gestures is feasible. The multiscale singular value PE eigenvalues describe the signal characteristics comprehensively, which lays the foundation for the successful classification of various gestures.

### 3.2. Classification Experiment

In this paper, the multiscale singular value PE eigenvalue of four muscles are input into the DBN for the classification experiment. In Figure 7, $\left(F_{1},F_{2},F_{3},F_{4}\right)$ are the inputs of the network, which represent the eigenvectors of ECR, ED, FDS and EPB muscles, respectively. $\left(Y_{1},Y_{2},\cdots ,Y_{9}\right)$ are the output of the network, representing nine kinds of gestures such as FFE, FFC,… and FTIF. The network structure of the DBN system directly affects the performance of the deep network, but there is no unified theoretical standard [35]. Thus, the experimental analysis is carried out respectively under the condition of the fixed hidden layer number and the hidden cell number.

The number of hidden layers in the DBN system has an important impact on the accuracy of the system. If the depth of the system is too deep, the parameter optimization will fall into local optimization, and if the depth is too shallow, the input features cannot be fully trained. Therefore, under the condition that the number of hidden cells in each layer is fixed at 200, the influence of the number of hidden layers on the accuracy is studied, as shown in Figure 12a. It can be seen from the figure that the accuracy does not always increase with an increase of system depth. When the depth of a hidden layer is 3, the accuracy of the DBN system reaches the maximum. With the increase of depth, the generalization ability of the DBN system is affected, and the over fitting phenomenon appears, which not only reduces the accuracy but also increases the training time.

The number of hidden cells plays an important role in the learning ability and training time of the system. In the DBN system, too many hidden cells will lead to overload, and the redundant cells will increase the complexity of the training process and reduce the overall accuracy. If the number of hidden cells is too small, the connection between the neurons will be reduced, thus ignoring the feature information and reducing the ability of feature learning and training. In this paper, a DBN system with a depth of 3 was selected to study the influence of hidden cell change on accuracy. It can be seen from Figure 12b that the accuracy does not increase with the number of hidden cells. When the number of hidden cell is 300, the input features are fully absorbed, and the accuracy reaches the maximum of 93%.

After the network structure of the DBN system is determined, the recognition accuracy is closely related to the number of training samples. The more training samples, the higher the accuracy, as shown in Figure 12c. In this paper, 20 groups of training samples of feature extraction experiment are used to train the DBN, and then 20 groups of test samples (180 cases) are used for gesture classification. In order to fully illustrate the advantages of the gesture recognition algorithm based on the combination of multiscale singular value PE and DBN, the gesture features are input into SVM, BP, DBN and CNN respectively for comparison. The results are shown in Table 3. In order to facilitate comparison and analysis, Figure 13 lists the correct recognition of the each gesture in detail under SVM, BP, DBN and CNN classifiers.

After research and analysis, it is found that although the calculation time of this method is moderate, the average accuracy is the highest (93.33%). Compared with CNN, the RBM optimization of each layer in the DBN system only achieves the optimal weight of its own layer, which improves the efficiency of the deep learning network by selecting an appropriate initial value. Compared with BP and SVM, the traditional shallow learning algorithm only absorbs the extracted features on the surface, and the accuracy of learning depends on the detail of feature extraction.

The DBN system constructed in this paper has little effect on the accuracy after the training of the optimal samples. It fully illustrates that the DBN system with its own characteristics of unsupervised learning and supervised fine-tuning, shows the advantages in deep data mining that traditional shallow learning does not have.

## 4. Conclusions

Gesture recognition is an important research direction of human-computer interaction. In this paper, a gesture recognition method based on multiscale singular value PE and DBN is discussed. Firstly, the four channel sEMG signals are collected in the process of gesture execution, and the time-frequency resolution of sEMG signals is improved by ST. Then, the multiscale SVD of the TFM is carried out, and the multiscale singular value PE is calculated as the eigenvalue. Gesture features are input into the DBN for classification, and nine kinds of gestures are recognized, with an average accuracy of 93.33%. The experimental results show that the multiscale singular value PE feature is especially suitable for pattern classification of DBNs. In addition, this method provides a certain reference value for bioelectrical signal processing.

At present, the research method in this paper is only tested on healthy people. In the next step, we will cooperate with the rehabilitation hospital to do further experiments and research, and test this method on a group of deaf mute patients receiving physical therapy. In future work, this method of gesture recognition will be coupled with the trajectory recognition system of gesture motion, and we will try to transplant it to the real-time system. Gesture recognition technology based on sEMG signals not only has important academic value, but also has a wide application prospect.

## Figures and Tables

**Figure 1 sensors-21-00119-f001:**
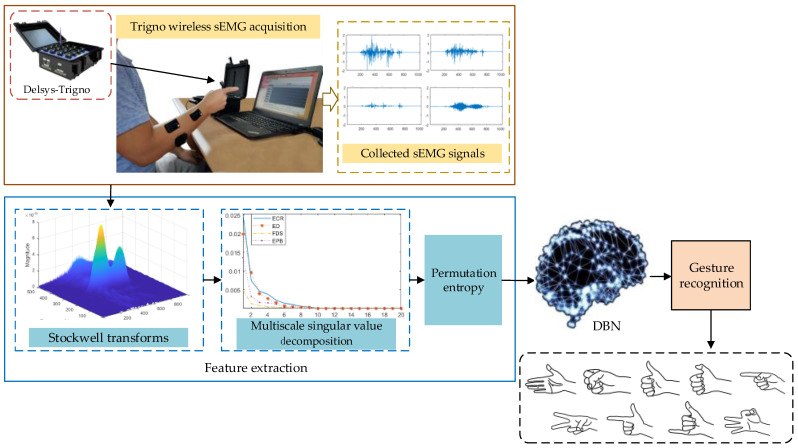
Framework of gesture recognition method.

**Figure 2 sensors-21-00119-f002:**
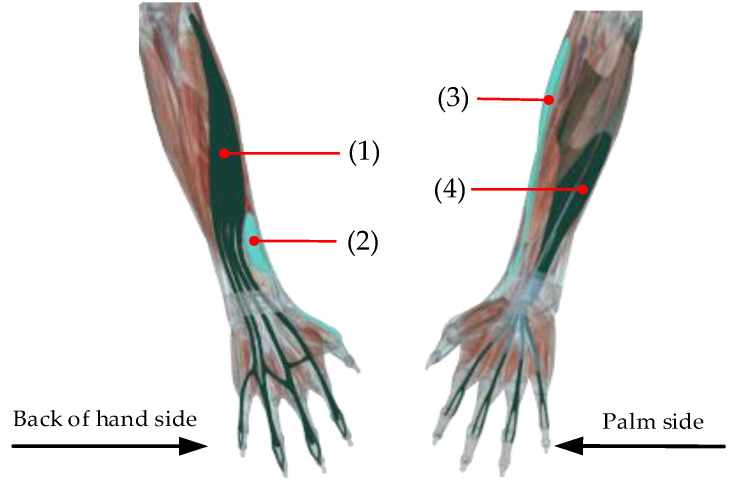
Position of surface electromyography (sEMG) signal acquisition sensors: (1) is the position of extensor digitorum (ED); (2) is position of extensor pollicis brevis (EPB); (3) is position of extensor carpi radius (ECR); and (4) is position of Flexor digitorum superficialis (FDS).

**Figure 3 sensors-21-00119-f003:**
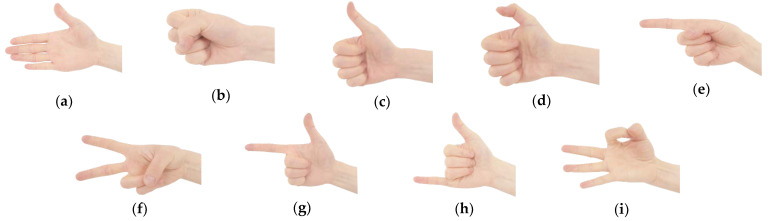
Schematic diagram of nine kinds of gestures: (**a**) five fingers extended; (**b**) five fingers closed; (**c**) extended thumb; (**d**) flexion of thumb; (**e**) extended index finger; (**f**) extended index and middle finger; (**g**) extended thumb and index finger; (**h**) extended thumb and pinkie; (**i**) flexion of thumb and index finger.

**Figure 4 sensors-21-00119-f004:**
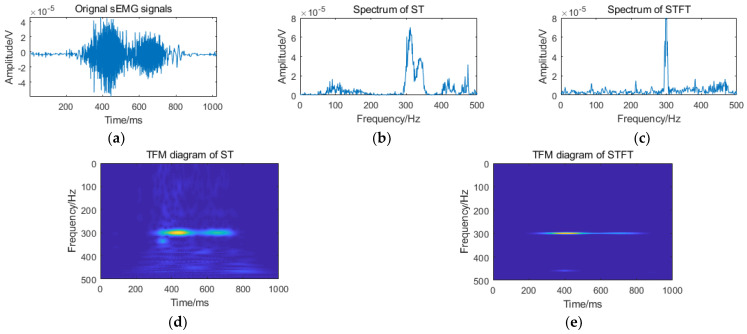
Comparison of the time-frequency diagram of the sEMG signal: (**a**) input sEMG signal; (**b**) spectrum of S-transform (ST); (**c**) spectrum of the short-time Fourier transform (STFT); (**d**) time-frequency matrix (TFM) diagram of ST; and (**e**) TFM diagram of STFT.

**Figure 5 sensors-21-00119-f005:**
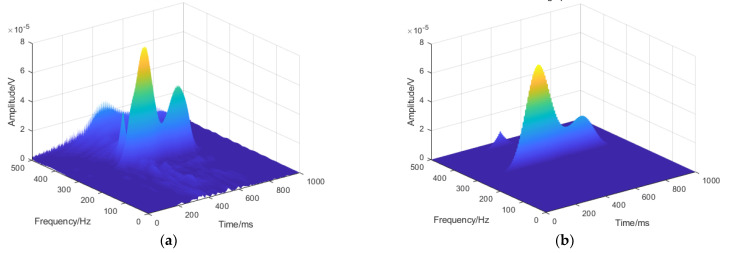
3D time-frequency-amplitude diagram of the sEMG signal: (**a**) 3D time-frequency-amplitude diagram of the ST and (**b**) 3D time-frequency-amplitude diagram of the STFT.

**Figure 6 sensors-21-00119-f006:**
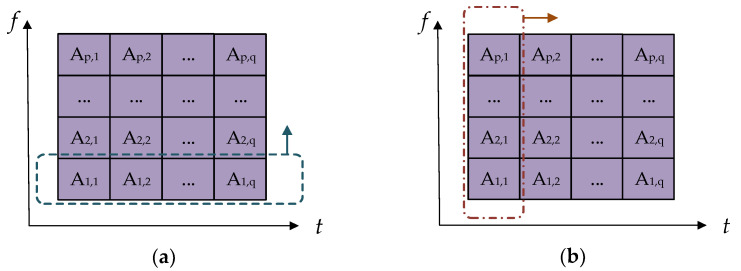
Schematic diagram of sub matrix division: (**a**) sub matrix divided along the frequency axis and (**b**) sub matrix divided along the time axis.

**Figure 7 sensors-21-00119-f007:**
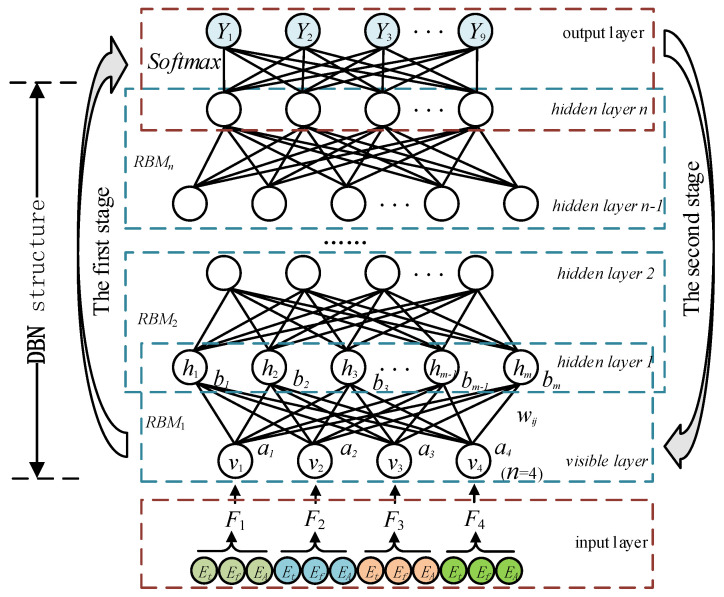
The DBN structure and its training process.

**Figure 8 sensors-21-00119-f008:**
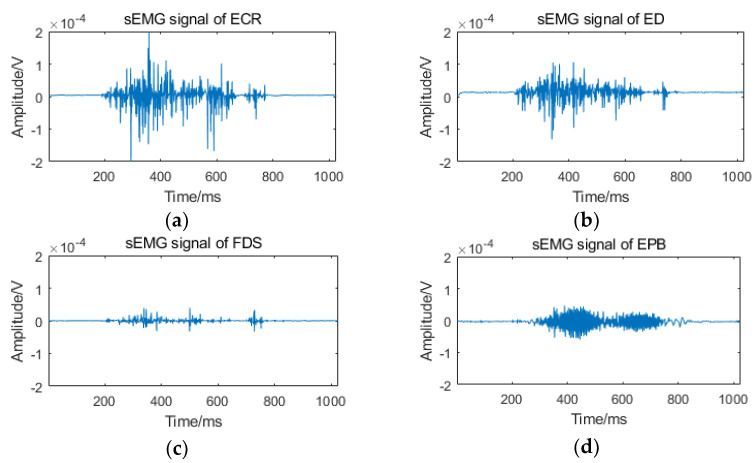
Collected original sEMG signals: (**a**) sEMG signal of ECR; (**b**) sEMG signal of ED; (**c**) sEMG signal of FDS; (**d**) sEMG signal of EPB.

**Figure 9 sensors-21-00119-f009:**
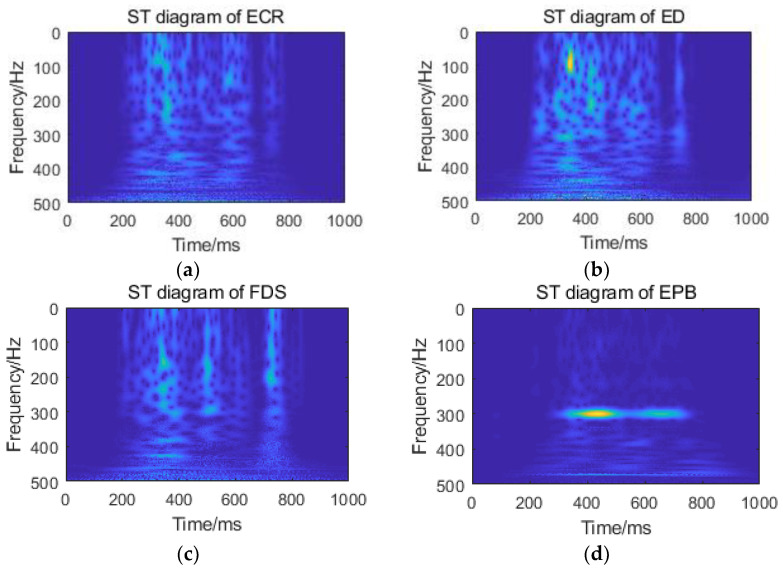
ST diagram of sEMG signals: (**a**) TFM diagram of ECR; (**b**) TFM diagram of ED; (**c**) TFM diagram of FDS; and (**d**) TFM diagram of EPB.

**Figure 10 sensors-21-00119-f010:**
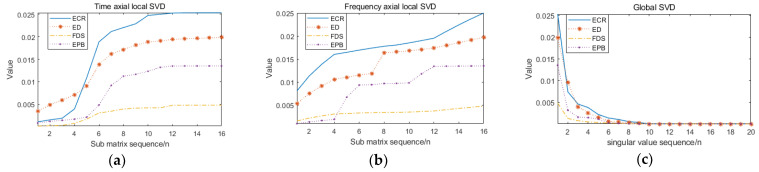
Multiscale singular value decomposition (SVD) analysis. (**a**) Maximum singular value of sub matric division along time axis; (**b**) Maximum singular value of sub matric division along frequency axis; (**c**) SVD of global matrix.

**Figure 11 sensors-21-00119-f011:**
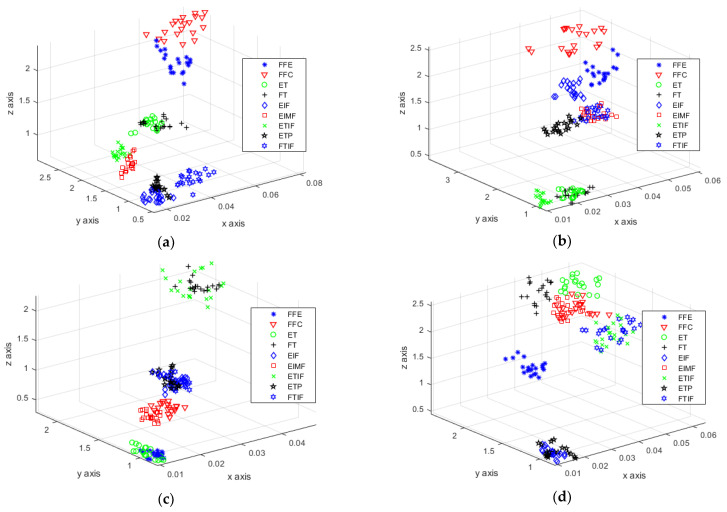
Multiscale singular value PE eigenvalue: (**a**) eigenvalue of ECR; (**b**) eigenvalue of ED; (**c**) eigenvalue of FDS; and (**d**) eigenvalue of EPB.

**Figure 12 sensors-21-00119-f012:**
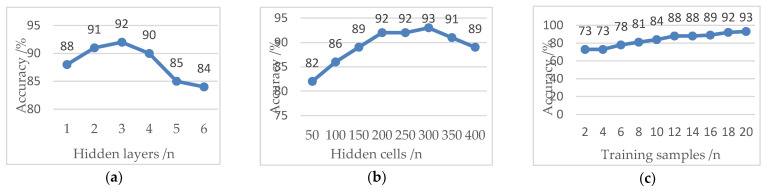
Accuracy of the deep belief network (DBN) system: (**a**) relationship between hidden layers and accuracy; (**b**) relationship between hidden cells and accuracy; and (**c**) relationship between training samples and accuracy.

**Figure 13 sensors-21-00119-f013:**
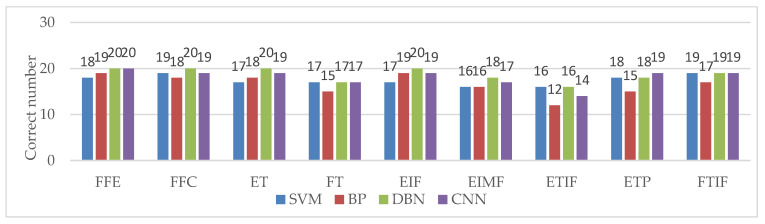
The correct recognition number of each gesture.

**Table 1 sensors-21-00119-t001:** Description of comparison of methods.

No.	Papers	Number of Channels	Types of Signal	Method	Accuracy
1	Reference [9]	4	sEMG, accelerometer, gyroscope	Optimized tree-structure classification	87.02%
2	Reference [10]	5	sEMG, accelerometer	Linear discriminant analysis	91.4%
3	Reference [11]	5	sEMG, accelerometer	Multi-stream hidden Markov models	90.2%
4	This paper	4	sEMG	Combination of multiscale singular value PE and DBN	93.33%

**Table 2 sensors-21-00119-t002:** Description of nine kinds of gestures.

No.	Name of Gesture Category	Abbreviation	Chinese SL Words ^1^	Example
1	Five fingers extended	FFE	Dajia ^2^; Xianzai ^3^; Kaixin ^4^	Figure 3a
2	Five fingers closed	FFC	Gongzuo ^5^; Jiayou ^6^	Figure 3b
3	Extend thumb	ET	Hao ^7^; Youyi ^8^	Figure 3c
4	Flexion of thumb	FT	Xiexie ^9^	Figure 3d
5	Extend index finger	EIF	Ni ^10^; Yi ^11^; Shuohua ^12^	Figure 3e
6	Extend index and middle finger	EIMF	Chang ^13^; Er ^14^	Figure 3f
7	Extend thumb and index finger	ETIF	Lv ^15^; Ba ^16^; Gong ^17^	Figure 3g
8	Extend thumb and pinkie	ETP	Dianhua ^18^; Qu ^19^; Liu ^20^	Figure 3h
9	Flexion of thumb and index finger	FTIF	Gui ^21^; Lianxi ^22^; San ^23^	Figure 3i

^1^ The Pinyin of Chinese characters, and will be explained here. ^2^ Everybody. ^3^ Now. ^4^ Happy. ^5^ Work. ^6^ Come on. ^7^ Well. ^8^ Friendship. ^9^ Thanks. ^10^ You. ^11^ One. ^12^ Talk. ^13^ Often. ^14^ Two. ^15^ Green. ^16^ Eight. ^17^ Public. ^18^ Phone. ^19^Go. ^20^ Six. ^21^ Expensive. ^22^ Contact. ^23^ Three.

**Table 3 sensors-21-00119-t003:** Comparison of accuracy of different classification methods.

Number	Feature Extraction	Classifier	Accuracy	Time Cost
1	Multiscale singular value PE	SVM	87.22% (157/180)	1.24 s
2	Multiscale singular value PE	BP	82.77% (149/180)	1.82 s
3	Multiscale singular value PE	DBN	93.33% (168/180)	2.17 s
4	Multiscale singular value PE	CNN	90.55% (163/180)	3.49 s

## Data Availability

The data used to support this study are available from the corresponding author upon request.

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
