# Peer review of "Gesture Recognition Based on Multiscale Singular Value Entropy and Deep Belief Network"

_sensors, 2020, doi:10.3390/s21010119_

Round 1
Reviewer 1 Report
Please see the attachment.

Reviewer 2 Report
The results presented in the manuscript "Gesture Recognition Based on Multiscale Singular Value Entropy and Deep Belief Network" are interesting and worth publishing in Sensors after a few modifications.
Please find my comments below.
1) It should be clearly written what is the novelty of the manuscript and what is the contribution of the authors in relation to other papers.
2) The organization of the paper should be included at the end of the Introduction section.
3) Please note that the symbols used in the text should be the same size and written in the same font as the main text.
4) References should be verified and carefully formatting in accordance with the guidelines of the Sensors.
Round 2
Reviewer 1 Report
Please see the attachment.
